# Exploratory study on the psychological impact of COVID-19 on the general Brazilian population

**Antonio P. Serafim**[1,2☯]*, **Ricardo S. S. Durães**[2☯], **Cristiana C. A. Rocca**[1☯], **Priscila D. Gonçalves**[1], **Fabiana Saffi**[1], **Alexandre Cappellozza**[3], **Mauro Paulino**[4], **Rodrigo Dumas-Diniz**[4], **Sofia Brissos**[5], **Rute Brites**[6], **Laura Alho**[7], **Francisco Lotufo-Neto**[1]

1 Department of Psychiatry, School of Medicine, University of São Paulo, São Paulo, Brazil, 2 Health Psychology Program, Methodist University of São Paulo, São Paulo, Brazil, 3 Social and Applied Sciences Center, Mackenzie Presbyterian University, São Paulo, Brazil, 4 Mind, Institute of Clinical and Forensic Psychology/Center for Research in Neuropsychology and Cognitive and Behavioral Intervention, Lisbon, Portugal, 5 National Legal Medicine Institute, Psychiatry and Clinical and Forensic Psychology, Lisbon Psychiatric Hospital, Lisbon, Portugal, 6 Department of Psychology, Autónoma University, Lisbon, Portugal, 7 Mind, Digital Human-Environment Interaction Lab, Universidade Lusófona de Humanidades e Tecnologias, Lisbon, Portugal

☯ These authors contributed equally to this work.
* a.serafim@hc.fm.usp.br

**Data Availability Statement:** The data are fully available and were made available on the submission platform in the topic

## Abstract

The COVID-19 pandemic has become one of the main international concerns regarding its impact on mental health. The present study aims to investigate the prevalence of depression, anxiety, and stress symptoms, and behavioral aspects amidst the COVID-19 pandemic in a Brazilian population. An online survey was administered from May 22 to June 5, 2020 using a questionnaire comprising of sociodemographic information, the Depression, Anxiety, and Stress Scale (DASS-21), and the Coping Strategies Inventory. Participants comprised 3,000 people from Brazil's 26 states and the Federal District, with an average age of 39.8 years, women (83%), married (50.6%), graduates (70.1%) and employees (46.7%). Some contracted the virus (6.4%) and had dead friends or relatives (22.7%). There was more consumption of drugs, tobacco, medication, and food (40.8%). Almost half of participants expressed symptoms of depression (46.4%), anxiety (39.7%), and stress (42.2%). These were higher in women, people without children, students, patients with chronic diseases, and people who had contact with others diagnosed with COVID-19. The existence of a group more vulnerable to situations with a high stress burden requires greater attention regarding mental health during and after the pandemic. That said, it should be emphasized that these findings are preliminary and portray a moment still being faced by many people amid the pandemic and quarantine measures. Therefore, we understand that the magnitude of the impacts on mental health will only be more specific with continuous studies after total relaxation of the quarantine.

"Supporting information" with the file "Covid-19 Research Database General Population Brazil".

**Funding:** This research received no external funding.

**Competing interests:** The authors declare no conflict of interest.

## Introduction

In a pandemic such as the COVID-19 outbreak, people tend to be more susceptible to physical, cognitive, behavioral, and emotional changes (which are not necessarily related to the clinical condition of the disease) [1]. The COVID-19 pandemic has become one of the main international concerns regarding its impact on mental health. Specific stressors related to the pandemic are affecting the general population, professionals working in direct patient care, and professionals who are not directly on the front line (configured as indirect trauma) [2, 3]. There is an increase on the demands relating to mental health, and moreover, there is the need to find ways to address the outbreak's consequences. The growing information on contamination and lethality rates associated with the absence of effective treatment or means of prevention such as vaccines may corroborate with the increase in demands on mental health [3, 4].

To flatten the virus's transmission curve and prevent a collapse in health systems, quarantines have been adopted as a public health measure in many places worldwide and often involve isolation and varying degrees of restrictions in people's movement. However, this form of protection has necessitated changes in people's lives such as in how schooling and work are conducted (e.g., homeschooling and home office, respectively) and also harms the economic sector due to the closing of most "non-essential" commercial stores [5, 6]. It has also necessitated changes in the habits of people, such as reducing social contact, adopting new standards of hygiene, and having flexibility regarding new forms of adaptation to the situation, since it requires a commitment to self-care and care in relation to others. At the same time, quarantines may also have an important impact on the mental health of the population, the extent of which, a priori, is unpredictable.

A study of cancer patients in isolation reported psychological problems such as anxiety, depression, sleep disorders, abstinence, regression, and hallucinations; it was also observed that children seemed to respond better to isolation [7]. Focusing on COVID-19, a study in China showed that more than half of 1,210 participants expressed some level of depression, anxiety, and stress [8].

In Lebanon, the imposition of quarantine measures was found to be related to Post Traumatic Stress Disorder (PTSD) symptoms during the second week of isolation, with symptoms worsening in the fourth week [9]. In more specific populations, such as university students in Spain, 2,530 students and workers of the University of Valladolid had moderate to extremely severe levels of symptoms of anxiety (21.3%), depression (34.2%), and stress (28.1%) [10].

As the COVID-19 crisis is permeated by more doubts than answers mainly in relation to effective treatments, its impact on mental health could be a continuous phenomenon, crossing the quarantine period, and thereby becoming a new clinical condition of the pandemic. Given the above, it is clear that the development of mental health care and psychosocial well-being programs is as necessary as physical health care. The Mental Health and Psychosocial Support for Staff (MHPSS) [11] have already highlighted the need for support and understanding programs as such programs will make coping more appropriate and safer for the population, which will certainly result in mitigating mental health problems.

As if the peculiar complexity of COVID-19 in Brazil, for example, territorial extension and great social inequality, was not enough, the pandemic emerged amid a political and scientific conflict, which notably divided federal management on one side and state and municipal management on the other regarding the understanding of quarantine as an indicated measure. This polarization puts in risk the actions to address the spread of COVID-19, which is mainly affecting people living in regions and houses that are in an increased vulnerability to contamination and dissemination.

During the survey (May 22-June 5, 2020), Brazil reached further than 820,000 confirmed cases and almost 40,000 deaths. According to Coronavirus Resource Center—Johns Hopkins University (https://coronavirus.jhu.edu/map.html), currently, five months after the first case of contagion registered (Jan 26) in the country, there are over 3,669,995 infected and almost 116,000 deaths for COVID-19. The first death was registered on March 12, 2020 (covid.saude. gov.br).

In light of all of this, this study aimed to verify the prevalence of symptoms of depression, anxiety, and stress, as well as the use of coping strategies and adaptation on behaviors amidst quarantine measures. This study also aimed to obtain general health data regarding COVID-19 in a Brazilian population. Our main hypothesis was that the pandemic situation of COVID-19 would be accompanied by the presence of symptoms of anxiety, depression, and stress, besides changes in the coping behavior of the general Brazilian population.

## Methods

### Participants and procedures

We developed an anonymous online questionnaire, using the Google Forms platform, which allowed us to reach people across the country. The research was disclosed on Facebook, Instagram, in the media sector of the Institute of Psychiatry of the Hospital das Clínicas at the University of São Paulo and of the Health Psychology program at the Methodist University of São Paulo. Cross-sectional data were collected between May 22 and June 5, 2020, covering the 26 Brazilian states and the Federal District to be filled by quarantined persons. The information on this study is posted for public access on "Plataforma Brasil" (https://plataformabrasil.saude. gov.br), which is the national and unified base of research records involving human beings linked to the National Health Council, an organ of the Ministry of Health.

This study used a quantitative cross-sectional design through non-probabilistic sampling using the "snowball" method. An online questionnaire was created via a Google Forms link; the link was also made available on social media to be filled out by people in quarantine. The inclusion criterion was being ≥18 years. Exclusion criteria included forms sent after June 5, 2020. Thus, a total of 3,031 anonymous participants answered the online survey and thirty-one was excluded once they were sent on after the June 5 deadline, and therefore, 3,000 participants with an average age of 39.8 (SD = 13.0) were included in this study and there was no missing data.

### Measures

We applied an online questionnaire consisting of 108 questions organized as follows: Twenty-one issues addressed to sociodemographic data (age, sex, marital status, Children, educational level, Number of people living at the residence), information about the general health conditions of the participants, about the COVID-19 contagion, death of relatives or friends, and behavior in the face of the demands generated by the pandemic. These alternatives were related to quantity formats (for example, the number of children), yes or no answers, or list of categories, such as the symptoms of COVID-19, which could be checked. To assess psychological symptoms, the Depression, Anxiety, and Stress Scale (DASS-21) was used, which had been adapted and validated in the Portuguese language [12]. This instrument consists of 21 questions, divided into three subscales, which are scored on a 4-point Likert scale. Each subscale is composed of seven items, which assess symptoms related to depression, anxiety, and stress. The total score is obtained by adding the scores of the seven items for each of the three subscales and multiplied by two.

The DASS-21 provides three scores (one per subscale), with possible scores having a minimum of "0" and a maximum of "21", expressing the levels normal, mild, moderate, severe, and extremely severe. Higher scores on each scale correspond to more negative affective states. According to the results of the DASS-21, we organized the sample into two groups: asymptomatic (normal level) and symptomatic (levels from mild to extremely severe). For the coping strategy data, we adapted the questions from the Coping Strategy Inventory [13] also to the online format, which assesses the efforts made by people in coping with stressful, chronic, or acute situations. It consists of 66 items that include thoughts and actions, and whose intensity is measured using a scale from 0 (not used) to 3 points (uses in large quantities).

The questionnaire was constructed in a way that did not allow closing the page without all the alternatives having been answered. Once it is an online survey and without the possibility of physical signature, all participants only accessed the online form after reading the Informed Consent and having accepted it by clicking on the button that configured the voluntary participation agreement in the survey.

## Statistical analysis

Data were analyzed using the Statistical Package for the Social Sciences version 23 (SPSS 23.0) software. Categorical variables were represented by frequency and percentage (%). The anxiety, depression, and stress scores did not present a normal distribution when the Kolmogorov-Smirnov normality test was performed ($p$-value = 0.001), resulting in the execution of non-parametric tests for later analyses. The Mann-Whitney U, Kruskal-Wallis H tests were used for consolidated comparison between demographic variables and symptomatic and asymptomatic groups and inter- and intra-group analyses. The Chi-square test was used to verify differences in the proportionality of frequencies to the analyzed variables and the respondent groups' (i.e., asymptomatic and symptomatic) depression, anxiety, and stress scores with a significance level of $p < 0.05$. The $p$-value $< 0.05$ was considered significant in this study for all statistical analysis. Spearman's correlation was used to analyze the association between variables, with the magnitude of the coefficients interpreted as either weak (0.1–0.3), moderate (0.4–0.6), or strong ($\geq 0.7$), both for positive and negative significance (cf. Dancey & Reidy [14]).

## Ethics procedures

All subjects gave their informed consent for inclusion before they participated in the study. This study was approved following the procedures of mental health research protocols for COVID-19 (SARS-COV-2) by the National Council of Ethics in Research (CONEP) of Brazil (CAAE: 30503920.5.0000.0008). As it was an online survey and without the possibility of a physical signature, the participants only had access to the form after reading and accepting the Free and Informed Consent Terms (ICF) that was available at the bottom of the online page.

## Results

Table 1 shows the sociodemographic characteristics and behavioral aspects of the participants.

Most respondents were women (83%; see Table 1). Ages ranged from 18 to 82 years, with 31 to 40 years (28.1%) being the most frequent age group. The criteria used for dividing the participants into age groups was based on Brazilian study by Pinheiro et al. [15] which analyzed stress during the COVID-19 pandemic in a situation of social distance. Most participants were married (50.6%), had college education (70.1%), and had children over 16 years old (24.4%). Most lived in houses of three to five people (61.1%), and most were either employed (46.7%) or self-employed (38.4%). A majority were reported to be in good health conditions (94.8%) and 25.9% reported chronic illness.

**Table 1. Sociodemographic and behavioral characteristics during the quarantine (N = 3000).**

| Variable | Categories description | f | (%) | M | (SD) |
|---|---|---|---|---|---|
| **Age** | | | | 39.8 | (13) |
| **Age group** | | | | | |
| | 18–20 years old | 140 | (4.7) | 19.1 | (0.8) |
| | 21–30 years old | 680 | (22.7) | 25.1 | (2.8) |
| | 31–40 years old | 843 | (28.1) | 36 | (2.8) |
| | 41–50 years old | 674 | (22.5) | 45.1 | (2.9) |
| | 51–60 years old | 435 | (14.5) | 55.1 | (2.8) |
| | 61–70 years old | 197 | (6.6) | 64.3 | (2.6) |
| | ≥71 (Until 82 years old) | 31 | (1.0) | 73.6 | (3) |
| **Sex** | | | | | |
| | Female | 2493 | (83) | | |
| | Male | 507 | (17) | | |
| **Status** | | | | | |
| | Single | 1156 | (38.5) | | |
| | Married | 1517 | (50.6) | | |
| | Divorced | 279 | (9.3) | | |
| | Widowed | 48 | (1.6) | | |
| **Children** | | | | | |
| | No children | 1422 | (47.4) | | |
| | ≤16 years old | 846 | (28.2) | | |
| | >16 years old | 732 | (24.4) | | |
| **Number of people living at the residence** | | | | | |
| | 1 or 2 | 1058 | (35.3) | | |
| | 03–05 | 1836 | (61.2) | | |
| | ≥6 | 106 | (3.5) | | |
| **Occupation** | | | | | |
| | Employed | 1402 | (46.7) | | |
| | Autonomous | 851 | (28.4) | | |
| | Unemployed | 278 | (9.3) | | |
| | Retired | 174 | (5.8) | | |
| | Student | 295 | (9.8) | | |
| **Educational level** | | | | | |
| | Primary and Secondary | 384 | (12.9) | | |
| | University graduate | 2103 | (70.1) | | |
| | Postgraduate (Master or PhD) | 330 | (17.0) | | |
| **Evaluation of current health status** | Normal/Very good | 2844 | (94.8) | | |
| | Weak | 753 | (25.1) | | |
| **Chronic disease** | Yes | 778 | (25.9) | | |
| **Close contact with infected people (Covid-19)** | Yes | 1055 | (35.2) | | |
| **Deaths of relatives or friends by Covid-19** | Yes | 682 | (22.7) | | |
| **Have done Covid-19 test** | Yes | 192 | (6.4) | | |
| | **Reason (symptoms)** | | | | |
| | Mild to Moderate | 179 | (93.3) | | |
| | Severe–ICU* | 13 | (6.7) | | |
| **Wears mask, regardless of the presence of symptoms** | Always | 2310 | (77.0) | | |
| **Agrees with the quarantine** | Yes | 2541 | (84.7) | | |
| **Time at home due to the pandemic** | 20-24hrs | 2086 | (69.5) | | |

*(Continued)*

**Table 1.** (Continued)

| Variable | Categories description | f | (%) | M | (SD) |
|---|---|---|---|---|---|
| **Slept more than normal** | From little to very much | 1839 | (61.3) | | |
| **Used drugs, medication, tobacco, or food for feeling good** | From little to very much | 1224 | (40.8) | | |

f = frequency, M = mean, SD = standard deviation.

*ICU = intensive care unit.

As for COVID-19, 35.2% reported contact with someone infected, 22.7% had relatives or friends who died, 6.4% contracted the virus, and 69.5% of the participants remained at home for 20 to 24 hours. Regarding the use of a mask and hygiene measures, 77 and 86.8%, respectively, reported always following these procedures. Regarding coping strategies, 61.3% reported an increase in hours of sleep, 40.8% perceived an increased intake of food, alcohol, drugs, tobacco, and medication.

Table 2 shows the results of the DASS-21 scale according to the sample distribution in two groups: "asymptomatic" (people who did not refer symptoms indicative scores for anxiety, depression, and stress) and "symptomatic" group (people who referred indicative scores of symptoms for anxiety, depression, and stress).

Table 2 shows that 46.4% of the participants showed symptoms of depression, 39.7% anxiety, and 42.2% stress. Analyzing the proportionality differences in frequencies in the levels of depression, anxiety, and stress, only the moderate level of stress was significantly more

**Table 2. Frequency of symptomatic and asymptomatic for depression, anxiety, and stress (N = 3000).**

| Variable | Level | f | (%) | p-value | Prevalence | f | (%) |
|---|---|---|---|---|---|---|---|
| **Depression (DASS-21‡)** | | | | | | | |
| Asymptomatic | | 1608 | (53.6) | | | | |
| Symptomatic | | 1392 | (46.4) | | | | |
| | Mild | 453 | (32.5) | 0.230+ | 18–30† | 137 | (21.2) |
| | Moderate | 489 | (35.1) | | 18–30† | 185 | (36.0) |
| | Severe/extremely | 450 | (32.1) | | 18–30† | 215 | (50.4) |
| **Anxiety (DASS-21‡)** | | | | | | | |
| Asymptomatic | | 1809 | (60.3) | | | | |
| Symptomatic | | 1191 | (39.7) | | | | |
| | Mild | 218 | (18.3) | 0.010+ | 18–30† | 64 | (31.7) |
| | Moderate | 493 | (41.4) | | 18–30† | 180 | (36.1) |
| | Severe/extremely | 480 | (40.3) | | 18–30† | 217 | (45.4) |
| **Stress (DASS-21‡)** | | | | | | | |
| Asymptomatic | | 1734 | (57.8) | | | | |
| Symptomatic | | 1266 | (42.2) | | | | |
| | Mild | 401 | (31.7) | 0.290+ | 31–40† | 127 | (30.7) |
| | Moderate | 422 | (33.3) | | 18–30† | 156 | (37.4) |
| | Severe/extremely | 443 | (35.0) | | 18–30† | 219 | (51.9) |

f = frequency.

+p-values from chi-squared test.

‡Depression, anxiety, and stress scale.

†Highest prevalence by age group.

**Table 3. Comparison of symptomatic and asymptomatic groups for depression.**

| Variable | Total sample (N = 3000) | | | | | Asymptomatic Depression | | | | Symptomatic Depression | | | | p(4) |
|---|---|---|---|---|---|---|---|---|---|---|---|---|---|---|
| | M | (SD) | Md | 95% CI | p(1) | f | (%) | M | p(2) | f | (%) | M | p(3) | |
| **Sex** | | | | | | | | | | | | | | |
| Female | 10.39 | (8.8) | 8 | 10.0–10.7 | 0.000[+] | 1277 | (42.6) | 4.7 | 0.000[+] | 1216 | (40.5) | 16.0 | 0.025[+] | 0.000[*] |
| Male | 7.5 | (8.5) | 4 | 6.8–8.2 | | 333 | (11.1) | 1.2 | | 174 | (6.0) | 15.2 | | |
| **Children** | | | | | | | | | | | | | | |
| No children | 11.6 | (9.3) | 10 | 11.1–12.1 | 0.000[+] | 630 | (21.0) | 5.4 | 0.000[+] | 792 | (26.4) | 16.5 | 0.016[+] | 0.000[*] |
| ≤16 years old | 9.2 | (8.3) | 6 | 8.6–9.8 | | 485 | (16.2) | 4.7 | | 361 | (12.0) | 15.2 | | |
| >16 years old | 7.5 | (7.7) | 6 | 6.9–8.0 | | 495 | (16.5) | 3.9 | | 237 | (8.0) | 14.9 | | |
| **Occupation** | | | | | | | | | | | | | | |
| Employed | 9.4 | (8.5) | 8 | 9.0–9.9 | 0.000[+] | 816 | (27.2) | 4.8 | 0.037[+] | 586 | (19.5) | 15.9 | 0.000[+] | 0.000[*] |
| Autonomous | 8.8 | (8.1) | 6 | 8.2–9.3 | | 488 | (16.0) | 4.6 | | 363 | (12.1) | 14.4 | | |
| Unemployed | 12.8 | (10.12) | 10 | 11.6–14.0 | | 99 | (3.3) | 4.1 | | 179 | (6.0) | 17.5 | | |
| Retired | 7.4 | (7.5) | 6 | 6.2–8.5 | | 118 | (3.9) | 4.3 | | 56 | (1.9) | 13.8 | | |
| Student | 14.1 | (10) | 12 | 13.0–15.3 | | 89 | (3.0) | 6.2 | | 206 | (6.9) | 17.6 | | |
| **Chronic disease** | | | | | | | | | | | | | | |
| Yes | 11.5 | (9.4) | 10 | 10.8–12.2 | 0.000[+] | 392 | (13.1) | 4.7 | 0.000[+] | 386 | (12.9) | 17.6 | 0.000[+] | 0.020[*] |
| No | 9.3 | (8.6) | 8 | 9.0–9.7 | | 1218 | (40.6) | 1.8 | | 1004 | (33.5) | 15.2 | | |
| **Close contact with infected people** | | | | | | | | | | | | | | |
| Yes | 12.4 | (9.9) | 10 | 11.6–13.3 | 0.000[+] | 241 | (8.0) | 4.7 | 0.000[+] | 261 | (8.8) | 18.5 | 0.000[+] | 0.004[*] |
| No | 9.4 | (8.5) | 8 | 9.1–9.7 | | 1369 | (45.6) | 1.9 | | 1129 | (37.6) | 15.3 | | |
| **Diagnostic of Covid-19** | | | | | | | | | | | | | | |
| Yes | 17.0 | (11.9) | 14 | 14.4–19.5 | 0.000[+] | 37 | (1.2) | 4.7 | 0.000[+] | 50 | (1.7) | 23.8 | 0.000[+] | 0.024[*] |
| No | 9.7 | (8.6) | 8 | 9.4–10.0 | | 1573 | (52.4) | 2.0 | | 1340 | (44.7) | 15.6 | | |

M = mean, Md = median, SD = standard deviation, f = frequency, CI = confidence interval.

[+]p(1), p(2), p(3) values from Mann-Whitney/Kruskall-Wallis.

[*]p(4) values from chi-squared test.

prevalent ($p < 0.01$, Chi-square). As for the distribution of symptoms according to age, participants between 18 and 30 years old showed higher frequencies regarding depression and anxiety with mild to extremely severe levels, and moderate to extremely severe concerning stress.

Tables 3, 4, and 5 show the analyses of the symptomatic and asymptomatic groups in relation to the variables gender, children, professional situation, presence of chronic disease, contact with people infected with COVID-19, and participants diagnosed with COVID-19.

As shown in Tables 3, 4, and 5, the levels of depressive, anxiety, and stress symptoms. Regarding to *p*-value for characteristics, it was for sex, children, occupation, and chronic disease 0.000 (depression) and 0.001 (anxiety and stress); close contact with infected people 0.000 (depression), 0.001 (anxiety), and 0.020 (stress); diagnostic of Covid 0.000 (depression), 0.001 (anxiety), and 0.008 (stress).

Concerning higher levels, it was similar group characteristic for both. For depression were higher in women (median 8; 95% CI 10.0–10.7), people without children (median 10; 95% CI 11.1–12.1), students (median 12; 95% CI 13.0–15.3), patients with chronic diseases (median 10; 95% CI 10.8–12.2), and people who had contact with others with COVID-19 (median 10; 95% CI 11.6–13.3) or properly diagnosed with the virus (median 14; 95% CI 14.4–19.5). Regarding to anxiety were higher in women (median 12; 95% CI 13.0–13.6), people without children (median 14; 95% CI 13.9–14.8), students (median 16; 95% CI 15.4–17.6), patients with chronic diseases (median 14; 95% CI 13.5–14.8), who had close contact with infected

**Table 4. Comparison of symptomatic and asymptomatic groups for anxiety.**

| Variable | Total sample (N = 3000) | | | | | Asymptomatic Anxiety | | | | Symptomatic Anxiety | | | | |
|---|---|---|---|---|---|---|---|---|---|---|---|---|---|---|
| | M | (SD) | Md | 95% CI | p(1) | f | (%) | M | p(2) | f | (%) | M | p(3) | p(4) |
| **Sex** | | | | | | | | | | | | | | |
| Female | 13.3 | (8.6) | 12 | 13.0–13.6 | 0.001[+] | 1445 | (48.2) | 8.3 | 0.420[+] | 1048 | (34.9) | 20.1 | 0.213[+] | 0.001[*] |
| Male | 9.7 | (8.5) | 8 | 8.9–10.4 | | 368 | (12.3) | 6.1 | | 139 | (4.6) | 19.3 | | |
| **Children** | | | | | | | | | | | | | | |
| No children | 14.3 | (9.1) | 14 | 13.9–14.8 | 0.001[+] | 747 | (24.9) | 8.5 | 0.001[+] | 675 | (22.5) | 20.7 | 0.001[+] | 0.001[*] |
| ≤16 years old | 12.2 | (8) | 12 | 11.7–12.8 | | 537 | (17.9) | 8.2 | | 309 | (10.3) | 19.3 | | |
| >16 years old | 10 | (7.9) | 8 | 9.5–10.6 | | 529 | (17.6) | 6.7 | | 203 | (6.8) | 18.6 | | |
| **Occupation** | | | | | | | | | | | | | | |
| Employed | 13.3 | (8.5) | 12 | 11.9–12.7 | 0.001[+] | 877 | (29.2) | 7.8 | 0.292[+] | 525 | (17.5) | 19.8 | 0.001[+] | 0.001[*] |
| Autonomous | 11.7 | (8) | 10 | 11.2–12.3 | | 563 | (18.8) | 8.1 | | 288 | (9.6) | 18.9 | | |
| Unemployed | 15.3 | (9.6) | 14 | 14.2–16.4 | | 125 | (4.2) | 8.1 | | 153 | (5.1) | 21.1 | | |
| Retired | 9.9 | (7.5) | 8 | 8.7–11.0 | | 127 | (4.2) | 7.0 | | 47 | (1.6) | 17.5 | | |
| Student | 16.5 | (9.7) | 16 | 15.4–17.6 | | 121 | (4.0) | 8.5 | | 174 | (5.8) | 22.1 | | |
| **Chronic disease** | | | | | | | | | | | | | | |
| Yes | 14.2 | (9.1) | 14 | 13.5–14.8 | 0.001[+] | 400 | (13.3) | 8.1 | 0.492[+] | 378 | (12.6) | 20.5 | 0.142[+] | 0.001[*] |
| No | 12.2 | (8.5) | 12 | 11.8–12.5 | | 1413 | (47.1) | 7.8 | | 809 | (27.0) | 19.7 | | |
| **Close contact with infected people** | | | | | | | | | | | | | | |
| Yes | 14.6 | (9.4) | 12 | 13.8–15.4 | 0.001[+] | 246 | (8.2) | 8.2 | 0.204[+] | 256 | (8.5) | 20.8 | 0.192[+] | 0.001[*] |
| No | 12.3 | (8.5) | 12 | 12.0–12.6 | | 1567 | (52.2) | 7.8 | | 931 | (31.0) | 19.8 | | |
| **Diagnostic of Covid-19** | | | | | | | | | | | | | | |
| Yes | 16.2 | (9.9) | 14 | 14.0–18.3 | 0.001[+] | 31 | (1.0) | 7.4 | 0.756[+] | 56 | (1.9) | 21 | 0.401[+] | 0.001[*] |
| No | 12.6 | (8.6) | 12 | 12.3–12.9 | | 1782 | (59.4) | 7.9 | | 1131 | (37.7) | 19.9 | | |

M = mean, Md = median, SD = standard deviation, f = frequency, CI = confidence interval.

[+]p(1), p(2), p(3) values from Mann-Whitney/Kruskall-Wallis.

[*]p(4) values from chi-squared test.

people (median 12; 95% CI 13.8–15.4) and participants diagnosed with the virus (median 14; 95% CI 14.0–18.3). Likewise, findings higher levels for stress were in women (median 14; 95% CI 14.4–15.0), people without children (median 14; 95% CI 15.9–16.8), students (median 18; 95% CI 18.3–20.3), patients with chronic diseases (median 14; 95% CI 14.7–15.9), and people who had contact with others with COVID-19 (median 14; 95% CI 14.5–16.0) and which were diagnosed with the virus (median 14; 95% CI 15.0–19.1). In the intragroup analyses, in relation to depression (Table 3), the prevalence of previous results is maintained ($p2 = 0.001$; $p3 = 0.001$).

Regarding anxiety (see Table 4) in the symptomatic group, only participants with or without children and different professional situations have significant differences, with couples without children and students presenting a higher level of anxiety symptoms ($p3 = 0.001$). For stress (see Table 5), there was no significant difference in stress levels in relation to gender ($p3 = 0.904$), presence of chronic disease ($p3 = 0.068$), occurrence of contact with people close to the virus ($p3 = 0.495$) or have a diagnosis of the disease ($p3 = 0.210$).

From the analysis of the frequency distribution between the asymptomatic and symptomatic groups, women, people without children, the unemployed, students, people with chronic diseases, and people who had contact with people who were diagnosed with the virus or who were diagnosed with COVID-19, expressed a higher proportion of depressive, anxious and

**Table 5. Comparison of symptomatic and asymptomatic groups for stress.**

| Variable | Total sample (N = 3000) | | | | | Asymptomatic Stress | | | | Symptomatic Stress | | | | p(4) |
|---|---|---|---|---|---|---|---|---|---|---|---|---|---|---|
| | M | (SD) | Md | 95% CI | p(1) | f | (%) | M | p(2) | f | (%) | M | p(3) | |
| **Sex** | | | | | | | | | | | | | | |
| Female | 14.7 | (8.1) | 14 | 14.4–15.0 | 0.001[+] | 1374 | (45.8) | 9.9 | 0.001[+] | 1119 | (37.3) | 20.6 | 0.904[+] | 0.001[*] |
| Male | 11.9 | (7.7) | 10 | 11.2–12.6 | | 363 | (12.1) | 8.5 | | 144 | (4.8) | 20.4 | | |
| **Children** | | | | | | | | | | | | | | |
| No children | 16.4 | (8.7) | 14 | 15.9–16.8 | 0.001[+] | 708 | (23.6) | 10.6 | 0.001[+] | 714 | (23.8) | 22.0 | 0.001[+] | 0.001[*] |
| ≤16 years old | 13.2 | (7) | 12 | 12.7–13.7 | | 504 | (16.8) | 9.5 | | 342 | (11.4) | 18.7 | | |
| >16 years old | 11.3 | (6.7) | 10 | 10.8–11.8 | | 525 | (17.5) | 8.4 | | 207 | (6.9) | 18.5 | | |
| **Occupation** | | | | | | | | | | | | | | |
| Employed | 13.3 | (7.6) | 12 | 12.9–13.7 | 0.001[+] | 841 | (28.0) | 9.1 | 0.001[+] | 561 | (18.7) | 19.5 | 0.001[+] | 0.001[*] |
| Autonomous | 13.5 | (7.3) | 12 | 13.0–14.0 | | 516 | (17.2) | 9.6 | | 335 | (11.2) | 19.4 | | |
| Unemployed | 18.1 | (9.4) | 16 | 17.0–19.2 | | 133 | (4.4) | 11.2 | | 145 | (4.8) | 24.3 | | |
| Retired | 11 | (5.7) | 10 | 10.1–11.8 | | 132 | (4.4) | 9.0 | | 42 | (1.4) | 17.3 | | |
| Student | 19.3 | (8.8) | 18 | 18.3–20.3 | | 115 | (3.8) | 12.5 | | 180 | (6.0) | 23.8 | | |
| **Chronic disease** | | | | | | | | | | | | | | |
| Yes | 15.3 | (8.6) | 14 | 14.7–15.9 | 0.001[+] | 396 | (13.2) | 9.5 | 0.947[+] | 382 | (12.7) | 21.2 | 0.068[+] | 0.001[*] |
| No | 13.9 | (7.8) | 12 | 13.6–14.2 | | 1341 | (44.7) | 9.7 | | 881 | (29.4) | 20.3 | | |
| **Close contact with infected people** | | | | | | | | | | | | | | |
| Yes | 15.2 | (8.9) | 14 | 14.5–16.0 | 0.020[+] | 257 | (8.6) | 9.6 | 0.756[+] | 245 | (8.2) | 21.1 | 0.495[+] | 0.001[*] |
| No | 14 | (7.9) | 12 | 13.7–14.3 | | 1480 | (49.3) | 9.6 | | 1018 | (33.9) | 20.4 | | |
| **Diagnostic of Covid-19** | | | | | | | | | | | | | | |
| Yes | 17 | (9.7) | 14 | 15.0–19.1 | 0.008[+] | 38 | (1.3) | 10.0 | 0.634[+] | 49 | (1.6) | 22.5 | 0.210[+] | 0.005[*] |
| No | 14.2 | (8) | 12 | 13.9–14.4 | | 1699 | (56.6) | 9.6 | | 1214 | (40.5) | 20.5 | | |

M = mean, Md = median, SD = standard deviation, f = frequency, CI = confidence interval.

[+]p(1), p(2), p(3) values from Mann-Whitney/Kruskall-Wallis.

[*]p(4) values from chi-squared test.

stress symptoms in relation to their own classification within the set of variables (p4 in Tables 3, 4, and 5). Table 6 presents the results of the correlation between studied variables.

Depression was shown to have a moderate positive result with anxiety and stress; weak positive with an occupation, sleeping more and using drugs, drugs or eating more to feel good; and moderate negative with age. Whereas anxiety was positive strong with stress; weak positive with chronic illness, COVID-19 diagnosis, sleeping more and using drugs, drugs or eating; weak negative with age and education level. Stress had a weak positive correlation with sleeping more and using some substance to feel better, and weak negative with age.

## Discussion

This study aimed to examine the psychological impact of COVID-19 in an adult sample in Brazil and had the participation of people from all regions of the country. Verifying the pandemic's impact on mental health may provide the needed information to promote policies to assist affected populations.

Our sample was characterized by more women and married people, ages ranged from 18 to 82 years, with those between 30 and 40 years having greater participation. Most had some college education, were working, and were in good health condition. A third of respondents disclosed that they had contact with people with COVID-19 and a small proportion contracted

**Table 6. Correlation of among sociodemographic variable, coping strategies, depression, anxiety, and stress (N = 3000).**

| Spearman's Correlation Coefficient | 1. | 2. | 3. | 4. | 5. | 6. | 7. | 8. | 9. | 10. | 11. | 12. |
|---|---|---|---|---|---|---|---|---|---|---|---|---|
| 1. Depression | | 0.607** | 0.680** | -0.301** | 0.132** | -0.095** | 0.053** | 0.048** | 0.064** | 0.091** | 0.289** | 0.352** |
| 2. Anxiety | | | 0.701** | -0.265** | 0.088** | -0.121** | 0.114** | 0.108** | 0.058** | 0.041* | 0.189** | 0.301** |
| 3. Stress | | | | -0.289** | 0.089** | -0.089** | 0.092** | 0.061** | 0.049** | 0.036* | 0.218** | 0.369** |
| 4. Age | | | | | -0.071** | 0.178** | 0.167** | -0.010 | -0.092** | -0.045* | -0.257** | -0.225** |
| 5. Occupation | | | | | | -0.191** | 0.029 | -0.011 | -0.024 | 0.202** | 0.103** | 0.037* |
| 6. Educational level | | | | | | | 0.021 | -0.036 | 0.073** | 0.065** | -0.070** | -0.008 |
| 7. Chronic disease | | | | | | | | -0.012 | -0.003 | 0.030 | 0.041* | 0.025 |
| 8. Diagnosis Covid-19 | | | | | | | | | -0.036* | -0.076** | 0.017 | -0.015 |
| 9. Agrees with the quarantine | | | | | | | | | | 0.193** | 0.067** | 0.037* |
| 10. Time at home | | | | | | | | | | | 0.110** | 0.024 |
| 11. Slept more than normal | | | | | | | | | | | | 0.246** |
| 12. Drugs, medication, or ate for feeling good | | | | | | | | | | | | |

*≤0.01.

**≤0.05.

the virus. The few studies exploring the pandemic's psychological impact had a similar prevalence of women [2, 8, 10, 16, 17], except for a study that investigated the psychological symptoms of ordinary Chinese citizens in the early stages of the pandemic, which had more men [18]. Women in many cultures assume overly broad responsibilities: besides their professional role, they often end up being at the heart of household chores and childcare [19]. Also, it must be noted that among women, there is a high prevalence of violence and abuse, which has been exhibited in other epidemics such as the Ebola and Zika virus outbreaks in 2013–2016 and 2015–2016, respectively. A field in which the majority of workers are predominantly women is that of health and social services, which necessitates the urgency to not only train women professionally but also to provide them with more resources when they assume primary responsibility for work domestic [19].

The sociodemographic variables observed in this study, such as the number of married people, educational level, and the number of residents in the house, are similar to those identified in a recent study in China [8]. However, for the participants in that study, the largest age group comprised those between 20 and 30 years and more than half were students.

## Psychological impact

Regarding psychological impact, symptoms of depression, anxiety, and mild to moderate stress, were found in almost half of our sample, which was in accordance with the presence of symptoms experienced less than a week before the conducting of the survey [12]. According to the DASS's foundation, the presence of depressive symptoms is associated with hopelessness, low self-esteem, and low incentive. So, as for anxiety is associated with physiological hyperstimulation and stress, which is an emotional state that varies according to an individual's assessment of situations experienced as a threat, damage, or challenge, coupled with increased irritability and limitation of frustration [18]. The presence of depressive, anxious, and stress symptoms may at some point cause emotional changes or physiological changes in the hypothalamic-pituitary-adrenal (HPA) axis, for example, as highlighted in past studies [20].

In this sense, one must consider that biological models have shown that hypo- sensitivity or hypersensitivity may be markers of "trait" of individuals with affective disorders and temperament characteristics similar to those found in patients with mood disorders (affective

temperament), have a higher probability of suffer from hopelessness and this is a predictive aspect for suicide risk [21, 22]. Thus, research that addresses behavioral signs related to anxiety, depression and mild to moderate stress associated with COVID-19 raise an important alarm for public health services, which need to be prepared for the increased demand and urgency in the treatment of new cases of mental disorders.

These findings corroborate the results of Wang et al. [8], who surveyed 210 Chinese people using the DASS-21 and IES-R (Impact of Event Scale-Revised) questionnaires. Their findings revealed severe levels of depressive symptoms (30.3%), anxiety (36.4%), and stress (32.1%). Our indexes were also of a serious level; however, they were higher on the DASS-21 scale compared to the study by Wang et al. [8]. The risks of the COVID-19 pandemic's psychological impact on the mental health of the general population have been discussed in previous studies, although these have been mostly theoretical in the form of editorials, letters to the editor, and comments [3, 4, 23–25].

Other studies exploring specific populations also show a psychological impact that corroborates our findings. A study assessing 460 Portuguese university students (with an average age of 20.14 years) using the DASS-21 observed high levels of anxiety, depression, and stress in this population [26]. In Spain, 2,530 students and workers of the University of Valladolid, presented moderate to extremely severe levels in terms of symptoms of anxiety (21.34%), depression (34.19%), and stress (28.14%) [10]. Another study, which evaluated psychopathological symptoms online through the Self-Report Symptom Inventory (SCL-90) in 1.600 participants, had more than 70% of participants expressing moderate to severe indices of anxiety, compulsions, and psychoticism [17]. Researchers in Lebanon using the Post-Traumatic Stress Disorder Checklist (PTSS) among 950 civilians, demonstrated that 33.2% of respondents had symptoms of PTSD.

As for depression, there was a higher prevalence in women than men in our study. Past studies have emphasized that depression is twice as prevalent in young women as in men, although it decreases throughout life and as triggers, women tend to have more internalizing symptoms [27]. Also, it is possible that regardless of age, professional status, marital status, education, and maternity, a condition such as a pandemic imposes an even greater burden on one's duties during confinement, causing greater psychological distress. These aspects also extend to stress anxiety levels.

It is well known that the pandemic has an emotional burden relating to concerns about contamination rates, people who need treatment in intensive care units, risk of death, and vulnerable populations [11]. Therefore, it was expected that participants who had contact with people close to or diagnosed with COVID-19 would express high levels of symptoms of depression, anxiety, and stress, which corroborates the findings of other studies [2, 8, 9, 28, 29].

Although the majority of the population works, has a degree, and is married, our results suggest that these variables are not protective factors for mental health and that the pandemic does indeed have an impact and magnitude that is difficult to measure. In the case of students, they probably feel more threatened and insecure about the future, in addition to having their mobility and social contact being reduced. In this population segment, research in Spain and Portugal showed indications of the pandemic's psychological impact [10, 26].

Despite the number of unemployed being less than 10% in our sample, there was an observed increase in psychological suffering. Although it was not verified whether unemployment resulted from the pandemic or not, the indicators of elevated symptoms of depression, anxiety, and stress corroborate previous findings on psychological processes and unemployment. Studies have highlighted that unemployed people were more likely to experience depressive symptoms, more stress, and less well-being compared to employed individuals [30, 31].

Also, unemployment is associated with an increased risk of mortality for those in the early and middle periods of their careers, compared to those at the end [32].

## Behavioral coping strategies

Coping strategy behaviors can be either positive or negative [33]. A little less than half of our sample reported an increase in sleeping hours, food intake, alcohol use, drug use, tobacco use, and medication because of the pandemic. These behaviors denote the participants' adoption of a negative escape-avoidance strategy, and such a strategy usually involves people imagining possible solutions to a problem without, however, taking actions to change them [33]. This finding corroborates with previous studies, since, in situations of high pressure and stress, people tend to adopt more negative strategies [34]. That said, the symptoms of stress may be associated with feelings of threat, harm, and challenge in the face of changes in behavior that the COVID-19 pandemic imposed on the population, aspects that have already been reported in past studies [35, 36].

The use of face masks and hand sanitizer was evidenced to not happen all the time, thus presenting a risk factor for contamination and the spread of the virus. This points out the need to develop programs that enable people to adopt healthy habits and behaviors, translated into concrete measures to reduce the contamination of COVID-19. In primary care, a method for prevention is through "counseling," which represents the directed use of communication and problem-solving techniques capable of stimulating people to change behaviors and adopt safer and healthier lifestyle habits. According to Oliveira et al. [37], the relevance of counseling for the promotion, prevention, and control of diseases is such that it is no longer possible to provide complete and technically correct primary care programs without the inclusion of counseling for healthy habits and behaviors.

In the scope of psychological impact, research thus far, even if still quite small, is quite suggestive of the possible damage of the pandemic to mental health, emphasizing the need for public health policies that value encouraging not only physical care but also emotional care, with programs that can stimulate and teach techniques for handling emotions, to reduce anxiety and stress [38]. In addition, it is necessary to consider that the occurrence of depressive, anxious, and high-stress symptoms in adults, which can also result from trait markers [21, 22], raises the concern that directing attention to children and adolescents is fundamental.

Children and adolescents depend on the care of protective figures, who are fragile and end up failing to provide the necessary support and care, leaving them in a situation of vulnerability and, therefore, very susceptible to stressful experiences [39].

## Study limitations

This study has limitations that must be considered when interpreting the results. As the sample composition was voluntary and online within a short period, we were unable to perform randomization, and so, for example, most participants were with a higher level of education, making the sample more selective. We also did not verify, in the cases of the unemployed, whether they lost their jobs during the pandemic. We also did not verify the presence of a previous history of mental disorders, which would certainly produce a clearer picture of the psychological impact in the face of the pandemic.

## Conclusion

Despite these limitations, our findings showed the presence of important psychological changes in the general population. Women, people without children, the unemployed, students, people with chronic diseases, and participants who had contact with close people

infected or diagnosed with the virus, were more susceptible to its psychological impact. This may suggest the existence of a group of greater vulnerability for situations with a high-stress load and which requires greater attention in terms of society's proposals for its management. The fact that women and people without children, as well as unemployed people, present higher levels of psychological changes raises the hypothesis that the situation of the pandemic may put them in a condition of lack of perspective, as the uncertainty about the future and other uncertainties tend to cause uncomfortable sensations, anguish, and anxiety, in addition to the possible feeling of helplessness. Therefore, expanding psychiatric care services, training more qualified professionals in the management of psychological impacts in the face of the pandemic, within the scope of cognitive-behavioral therapy and psychoeducation, for example, will certainly reduce the mental health problems in this population.

That said, it should be emphasized that these findings are preliminary and portray a moment still being faced by many people amid the pandemic and quarantine measures. Therefore, we understand that the magnitude of the impacts on mental health will only be more specific with continuous studies after total relaxation of the quarantine.

## Supporting information

**S1 Data.**
(XLS)

## Acknowledgments

We would like to thank all the people who were willing to answer this survey.

## Author Contributions

**Conceptualization:** Antonio P. Serafim, Ricardo S. S. Durães, Mauro Paulino, Sofia Brissos, Rute Brites, Laura Alho, Francisco Lotufo-Neto.

**Data curation:** Antonio P. Serafim, Ricardo S. S. Durães, Cristiana C. A. Rocca, Priscila D. Gonçalves, Fabiana Saffi, Alexandre Cappellozza, Rodrigo Dumas-Diniz.

**Formal analysis:** Antonio P. Serafim, Cristiana C. A. Rocca, Priscila D. Gonçalves, Fabiana Saffi, Alexandre Cappellozza, Francisco Lotufo-Neto.

**Investigation:** Antonio P. Serafim, Ricardo S. S. Durães, Cristiana C. A. Rocca, Fabiana Saffi.

**Methodology:** Antonio P. Serafim, Ricardo S. S. Durães, Cristiana C. A. Rocca, Alexandre Cappellozza, Mauro Paulino, Rodrigo Dumas-Diniz.

**Project administration:** Antonio P. Serafim.

**Writing – original draft:** Antonio P. Serafim, Ricardo S. S. Durães, Cristiana C. A. Rocca, Priscila D. Gonçalves, Francisco Lotufo-Neto.

**Writing – review & editing:** Antonio P. Serafim, Ricardo S. S. Durães, Cristiana C. A. Rocca, Priscila D. Gonçalves, Fabiana Saffi, Francisco Lotufo-Neto.

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
