## [Decision Letter · Decision Letter 0]

27 Nov 2020

PONE-D-20-26835

Exploratory study on the psychological impact of COVID-19 on the general Brazilian population

PLOS ONE

Dear Dr. Serafim,

Thank you for submitting your manuscript to PLOS ONE. After careful consideration, we feel that it has merit but does not fully meet PLOS ONE’s publication criteria as it currently stands. Therefore, we invite you to submit a revised version of the manuscript that addresses the points raised during the review process.

Please follow closely suggestions for revisions by reviewers 1 and 2. Reviewer 1 requests more information about the reporting of exclusion criteria as well as on the online questionnaire. Reviewer 1 also recommends upfronting the most relevant study findings at the beginning of the discussion. In addition, please reference the issue of the unique sensory processing patterns in subjects with anxiety and depression and refer to the potential risk of suicide, as discussed by reviewer 1.

Please address the issues reviewer 2 raises with respect to statistical analysis, especially points 3-13. If there were any leaks to the sample, how would the authors repair them. Please also revise your central hypothesis presented at the end of the introduction following reviewer 2' suggestions. Please also address question 2 and check out The Equator Network.

We look forward to receiving your revised manuscript.

Kind regards,

M. Harvey Brenner, PhD

Academic Editor

PLOS ONE

Journal Requirements:

3. Please include your tables as part of your main manuscript and remove the individual files. Please note that supplementary tables (should remain/ be uploaded) as separate "supporting information" files

4. Please ensure that you include a title page within your main document. You should list all authors and all affiliations as per our author instructions and clearly indicate the corresponding author.

5.We note that you have indicated that data from this study are available upon request. PLOS only allows data to be available upon request if there are legal or ethical restrictions on sharing data publicly. For more information on unacceptable data access restrictions, please see http://journals.plos.org/plosone/s/data-availability#loc-unacceptable-data-access-restrictions.

Reviewers' comments:

Reviewer's Responses to Questions

**Comments to the Author**

1. Is the manuscript technically sound, and do the data support the conclusions?

Reviewer #1: Yes

Reviewer #2: Yes

2. Has the statistical analysis been performed appropriately and rigorously? 

Reviewer #1: Yes

Reviewer #2: Yes

3. Have the authors made all data underlying the findings in their manuscript fully available?

Reviewer #1: Yes

Reviewer #2: Yes

4. Is the manuscript presented in an intelligible fashion and written in standard English?

Reviewer #1: Yes

Reviewer #2: Yes

5. Review Comments to the Author

Reviewer #1: This is, in summary, an interesting paper aimed to explore the prevalence of depression, anxiety, and stress symptoms, and behavioral aspects amidst the COVID-19 pandemic in a Brazilian population. The authors reported that some individuals contracted the virus and had dead friends or relatives. In addition, there was more consumption of drugs, tobacco, medication, and food. Importantly, almost half of participants, especially females, subjects without children, students, patients with chronic diseases, and individuals who had contact with others diagnosed with COVID-19 expressed symptoms of depression, anxiety, and stress.

The authors may find as follows my main comments/suggestions.

First, when within the Introduction section, the authors correctly reported that isolation and quarantine linked to Covid-19 infection are associated with psychological problems such as anxiety, depression, sleep disorders, abstinence, regression, and hallucinations, they could even briefly describe the involvement of sensory perception whch is implicated in emotional processes of individuals with psychological problems. Importantly, the unique sensory processing patterns in subjects psychological problems have been reported. Hyposensitivity or hypersensitivity may be "trait" markers of individuals with anxiety and depression and interventions should refer to the individual unique sensory profiles and their behavioral and functional impact in the context of real life. Thus, given the above information, my suggestion is to include throughout the manuscript, the paper published in 2016 on Psychiatry Res (PMID: 26738981).

In addition, the authors might even report the link between anxiety, depression, and specific adverse clinical outcomes (e.g., suicidal behavior). Importantly, depressed patients may frequenly display affective dysregulated temperaments and suicidal behavior. Overall, more than half of patients with dysregulated temperaments reported higher levels of hopelessness and may be considered at suicide risk. Thus, in order to briefly address the mentioned topics (although i understand that the link between anxiety, depression, and negative outcomes including suicidal behavior) is not the main aim of investigation of the present manuscript, the study below (PMID: 23104655) could be included throughout the main text by the authors.

Moreover, as the most relevant aims/objectives of the present study have been directly reported throughout the Introduction section, the main study hypotheses should be similarly described within the main text.

Furthermore, as inclusion criteria have been well described, exclusion criteria need to be extensively reported.

Furthermore, more information about the online questionnaire comprising 108 questions could be provided for the general readership.

In addition, the authors should immediately present and discuss, in the first lines of the Discussion section, their most relevant study findings. Conversely, they seem to focus with redundance on the main introductive statements as well as upon the main aims/objectives of this paper that should be stressed elsewhere within the main text.

Finally, which type of interventions the authors suggested for subgroups of subjects who were found to be more susceptible to the psychological impact of Covid-19 infection? Here, more details/ information are required to this specific regard.

Reviewer #2: Dears author,

The review of the article allows us to verify the commitment and the accuracy of the authors. So this review is intended to indicate some minor modifications listed below:

1- I suggest the revision of the central hypothesis presented at the end of the introduction. Although the authors compare the findings with other studies, the hypothesis does not highlight the main findings of the present study.

2 - It is unclear whether the authors make use of guidelines for scientific disclosure and transparency. Considering the methodological design and the way that the data were obtained, the use of these guidelines is encouraged (eg CHERRIES, STROBE ...). Therefore, authors can check out The EQUATOR Network.

3 - it is unclear if there were any leaks to the sample.

4 - it is unclear whether there are any missing in the data, and how these were considered for statistical analysis.

5 - I suggest improving the information regarding the socio-demographic questionnaire. With more specificity on questions and answer categories.

6 - It is not clear how the authors made the adaptation of the "Coping Strategy Inventory" and the criteria used by them.

7 - The last paragraph of item 2.1 would be better allocated in item 2.3.

8 - In the ‘data analysis’, it is not clear whether the significance level adopted concerns only chi-square or all statistical test.

9 - It is encouraged to refer to the literature, regarding the parameters adopted for the interpretation of the Rho.

10 - The criteria used for dividing the participants into age groups are not clear.

11- Table 1 reports data that were not used in other statistical analyzes and that contribute little to the discussion of the study. Authors are invited to verify the need for these data in the article.

12 - The inclusion of subtitles in table 2 is suggested, regarding the statistical test used to provide the p value.

13 - In table 3, the mean and standard deviation values were provided. However, considering the non-parametric analyzes, the inclusion of the median, respective CI or Interquartile Range, plus the percentage of the frequency for the chi-square test is suggested.

6. PLOS authors have the option to publish the peer review history of their article (what does this mean?). If published, this will include your full peer review and any attached files.

Reviewer #1: No

Reviewer #2: No

---

## [Author Response · Author response to Decision Letter 0]

22 Dec 2020

We appreciate the opportunity to review and resubmit our manuscript PONE-D-20-26835 – “Exploratory study on the psychological impact of COVID-19 on the general Brazilian population” in accordance to the reviewers´ suggestions. 

In this way, we review the manuscript and respond to all concerns of reviewer 1 and 2. Modifications to the manuscript are highlighted in yellow

---

## [Decision Letter · Decision Letter 1]

11 Jan 2021

Exploratory study on the psychological impact of COVID-19 on the general Brazilian population

PONE-D-20-26835R1

Dear Dr. Serafim,

We’re pleased to inform you that your manuscript has been judged scientifically suitable for publication and will be formally accepted for publication once it meets all outstanding technical requirements.

Kind regards,

M. Harvey Brenner, PhD

Academic Editor

PLOS ONE

Additional Editor Comments (optional):

Reviewers' comments:

Reviewer's Responses to Questions

**Comments to the Author**

1. If the authors have adequately addressed your comments raised in a previous round of review and you feel that this manuscript is now acceptable for publication, you may indicate that here to bypass the “Comments to the Author” section, enter your conflict of interest statement in the “Confidential to Editor” section, and submit your "Accept" recommendation.

Reviewer #1: All comments have been addressed

Reviewer #2: All comments have been addressed

2. Is the manuscript technically sound, and do the data support the conclusions?

Reviewer #1: Yes

Reviewer #2: Yes

3. Has the statistical analysis been performed appropriately and rigorously? 

Reviewer #1: Yes

Reviewer #2: Yes

4. Have the authors made all data underlying the findings in their manuscript fully available?

Reviewer #1: Yes

Reviewer #2: Yes

5. Is the manuscript presented in an intelligible fashion and written in standard English?

Reviewer #1: Yes

Reviewer #2: Yes

6. Review Comments to the Author

Reviewer #1: In the revised paper, the authors addressed most of the major questions raised by Reviewers improving both the main structure and quality of the present paper. I have no further additional comments.

Reviewer #2: In this review, I was able to verify the attention and care of the authors in answering my previous comments. I don't have new recommendations.

7. PLOS authors have the option to publish the peer review history of their article (what does this mean?). If published, this will include your full peer review and any attached files.

Reviewer #1: No

Reviewer #2: No

---

## [Editor Report · Acceptance letter]

13 Jan 2021

PONE-D-20-26835R1 

Exploratory study on the psychological impact of COVID-19 on the general Brazilian population 

Dear Dr. Serafim:

I'm pleased to inform you that your manuscript has been deemed suitable for publication in PLOS ONE. Congratulations! Your manuscript is now with our production department. 

Kind regards, 

on behalf of

Professor M. Harvey Brenner 

Academic Editor

PLOS ONE